# Epicardial conduction abnormalities in patients with Arrhythmogenic Right Ventricular Cardiomyopathy (ARVC) and mutation positive healthy family members – A study using electrocardiographic imaging

**Varvara Kommata[1]\*, Elena Sciaraffia[1], Carina Blomström-Lundqvist[1,2]**

**1** Department of Medical Sciences, Uppsala University, Uppsala, Sweden, **2** Department of Cardiology, School of Medical Sciences, Faculty of Medicine and Health, Örebro University, Örebro, Sweden

\* barbara.kommata@gmail.com

**Data Availability Statement:** Data contains extra sensitive information and cannot be shared publicly

## Abstract

### Background

The diagnosis of arrhythmogenic right ventricular cardiomyopathy (ARVC) in early stages is challenging. The aim of this study was therefore to investigate whether electrocardiographic imaging (ECGI) can detect epicardial conduction changes in ARVC patients and healthy mutation-carriers (M-carriers).

### Method

Twelve ARVC patients, 20 M-carriers and 8 controls underwent 12-lead ECG, signal-averaged ECG, 2-dimensional echocardiography, 24-hours Holter monitoring and ECGI (body surface mapping and computer tomography with offline analysis of reconstructed epicardial signals). Total and Right Ventricular Activation Time (tVAT and RVAT respectively), area of Ventricular Activation during the terminal 20 milliseconds ($aVAte_{20}$) and the activation patterns were compared between groups.

### Results

In ARVC patients the locations of $aVAte_{20}$ were scattered or limited to smaller parts of the right ventricle (RV) versus in controls, in whom $aVAte_{20}$ was confined to right ventricular outflow tract (RVOT) and left ventricle (LV) base (+/- RV base). ARVC patients had smaller $aVAte_{20}$ ($35cm^2$ vs $87cm^2$, $p<0.05$), longer tVAT (99msec vs 58msec, $p<0.05$) and longer RVAT (66msec vs 43msec, $p<0.05$) versus controls. In 10 M-carriers (50%), the locations of $aVAte_{20}$ were also eccentric. This sub-group presented smaller $aVAte_{20}$ ($53cm^2$ vs $87cm^2$, $p = 0.009$), longer RVAT (55msec vs 48msec, $p = 0.043$), but similar tVAT (65msec vs 60msec, $p = 0.529$) compared with the M-carriers with normal activation pattern.

without permission from the Ethics Review Board. Permission can be obtained from the Swedish Ethical Review Authority (registrator@etikprovning. se, adress: Swedish Ethical Review Authority, Box 2110, 750 02 Uppsala) upon request, for researchers who meet the criteria for access to confidential data.

**Funding:** The study received unrestricted grants from the Swedish Heart-Lung Foundation, Medtronic and Selanders Stiftelse. The funders had no role in study design, data collection and analysis, decision to publish, or preparation of the manuscript.

**Competing interests:** Due to ethical considerations, the data are available upon request. This does not alter, though, our adherence to PLOS ONE policies on sharing data and materials.

## Conclusions

ECGI can detect epicardial conduction abnormalities in ARVC patients. Moreover, the observation of localized delayed RV epicardial conduction in M-carriers suggests an early stage of ARVC and may be a useful diagnostic marker enhancing an early detection of the disease.

## Introduction

The diagnosis of Arrhythmogenic Right Ventricular Cardiomyopathy (ARVC), which predominantly affects the right ventricle (RV), is often challenging, particularly in its early stages [1]. Sudden cardiac death (SCD), the most devastating manifestation of the disease, is uncommon but can afflict young victims during the first stages of ARVC [2]. An early detection and adequate diagnosis of the disease is therefore crucial, which unfortunately is still complex, particularly in families without a known pathogenic mutation [3].

The fibrofatty replacement of the myocardium initiates in the epicardium or midmyocardium and extends to the endocardium resulting in transmural lesions [4]. Epicardial abnormalities may be the earliest signs of ARVC and have been reported to be more extensive compared with endocardial abnormalities [5]. Although these findings particularly support the use of epicardial mapping for early detection of the disease, such mapping systems are invasive and less suitable for repeated evaluations of apparently healthy individuals on long-term.

Electrocardiographic imaging (ECGI) using body surface recordings from 252 leads is a non-invasive multiple-leads based mapping system developed during the last decades for the reconstruction of epicardial signals [6]. Although initially used for diagnosis of atrial arrhythmias, its application in ARVC has also been evaluated demonstrating areas with delayed conduction, fractionated electrograms, and repolarisation abnormalities within regions harbouring ventricular ectopies [7, 8].

We have previously reported about repolarisation and depolarisation abnormalities in ARVC patients and gene carriers using a body surface mapping system [9, 10]. Given the observation that epicardial abnormalities are more extensive in ARVC patients, we hypothesised that the epicardium would be affected earlier than other regions and that such changes may be detected by non-invasive epicardial mapping for diagnostic purposes and potentially even be of value as prognostic markers [5].

In the present study the aim was to evaluate whether ECGI could detect epicardial depolarisation abnormalities in ARVC patients and potentially also unmask such changes in genetically predisposed individuals without morphological or clinical signs of the disease.

## Methods

### Study population

The present cross-sectional study was conducted at Uppsala University Hospital between December 2018 and April 2019. Individuals were prospectively included in three study groups; an ARVC group with a definite diagnosis according to 2010 Task Force Criteria (TFC), a mutation (M)-carrier group who were their relatives without structural abnormalities or ventricular arrhythmias but positive for the familial ARVC desmosomal mutation, and a control group of relatives who had tested negative for the familial desmosomal mutation [3]. The inclusion and exclusion criteria have been described in detail in previous publications [9, 10].

The diagnostic evaluations for ARVC included resting 12-leads ECG, Signal Average ECG (SAECG), 24-hour Holter monitoring and 2-dimensional (D)-echocardiography with standardized right ventricular projections in all study subjects.

The study was approved by the Regional Ethical Review Board (Dnr2018/369) and complied with the Declaration of Helsinki.

## Non-invasive electrocardiographic imaging (ECGI)

The ECGI methodology for the reconstruction of epicardial signals has been previously described [6, 11, 12]. Body surface ECG recordings were performed with CardioInsight non-invasive Electrocardiographic Mapping (ECM) System, software version 3.1, using a 252 electrode vest (CardioInsight[TM], Medtronic, MN). A low radiation CT scan was performed in order to visualize the vest electrodes in relation to the epicardium. Using the heart geometry recreated by the CT scan, the recorded body surface signals and the algorithm of the inverse solution method, 3-dimensional maps of the heart displaying the unipolar epicardial electrograms were reconstructed automatically by the system.

**Offline signal analysis.** The data were analyzed offline with CardioInsight[TM] non-invasive Electrocardiographic Mapping (ECM) System, software version 3.1. For additional signal analysis, the commercial software was extended using a customized offline analysis tool in Matlab [13].

For each study subject, three random beats were averaged automatically by the system from at least 200 beats, in order to minimize noise. Activation maps for each one of the three averaged beat were then created and analyzed offline. The local epicardial activation, automatically annotated by the system, was defined as the latest negative $dV/dt$ before the T-wave of the local electrogram. The annotations were manually reviewed by the first author and adjusted whenever needed. The anatomic locations of the earliest and latest local epicardial activations of the ventricles were defined.

The terminal ventricular activation was defined as the epicardial activation during the terminal 20 milliseconds (msec). The following variables were defined, measured and calculated;

- tVAT; the total Ventricular Activation Time, defined as the time from the earliest activation to the latest ventricular activation.

- $aVAte_{20}$; the area of the terminal Ventricular Activation, which was measured in order to evaluate the epicardial conduction during the terminal ventricular activation.

- $TaVAte_{20}$; the mean time during which all areas within $aVAte_{20}$ had completed their terminal activation.

The tVAT, $aVAte_{20}$ and $TaVAte_{20}$ were measured automatically by the system for each of the three beats separately and the mean values of the three measurements were calculated for comparison between the three study groups.

The morphology and timing of the reconstructed epicardial signals during the total RV activation were studied separately. For this purpose, the "Activation Time Editing" tool of the CardioInsight[TM] software was used. The earliest and latest activation of the RV were detected by manually scanning the epicardial signals using the multiple electrodes of the Activation Time Editing tool. The following variables were defined and calculated;

- RVAT; the Right Ventricular Activation Time, defined as the time from the earliest local RV activation to the latest local RV activation,

- RVAT-AW; the activation time for the RV anterior wall

- RVAT-IW; the activation time for the RV inferior wall.

The RV epicardial signals were classified as having an rS, Rs, rSr', qRs or qR morphology. A fractionated electrogram was defined as having at least two negative deflections separated by at least one positive deflection, provided that the amplitude of the smallest negative deflection was at least 10% of the amplitude of the deepest negative deflection, in order to exclude noisy signals.

The T-waves were defined as positive or negative if the deflection of the T-wave was above or below the baseline, respectively, and as isoelectric or biphasic if the T-wave was flat or had both a positive and negative deflection.

**Anatomical definitions.** The borders between the right ventricle (RV) and left ventricle (LV) were defined by i) the left anterior descending (LAD) coronary artery in the interventricular sulcus of the RV anterior wall (RVAW), ii) the posterior interventricular sulcus of the RV inferior wall (RVIW) and iii) confirmed by an abrupt change of the vector of the epicardial signals.

The RVAW was defined as the anterior part of the RV extending from the base to the apex. The right ventricular outflow tract (RVOT) was defined as the area just below the pulmonary valve extending to the tricuspid annulus on the RV base and the RV free wall. The RV free wall (RVFW) was defined as the mid part of the RVAW between the base and the RV apex. The RV antero-paraseptal region was defined as the area adjacent to the LAD. The RVIW consisted of the diaphragmatic surface of the RV and was divided into a basal and apical segment [14, 15].

## Statistical analysis

All statistical analyses were performed in SPSS statistical software (IBM SPSS statistics, version 28). Continuous variables were presented as mean ± standard deviation (SD) and categorical variables as percentages. The normal distribution of the data was tested with the Shapiro-Wilk test. Comparisons of continuous variables were performed using Mann-Whitney U test or Kruskal-Wallis test and comparisons of categorical variables with Pearson chi-square test. Correction for multiple comparisons was performed when appropriate (inclusively in single pairwise comparisons). A p-value of $\leq 0.05$ was regarded as statistically significant.

## Results

### Patients

The study cohort included 8 controls, 12 ARVC patients and 20 M-carriers, previously described in detail [9, 10]. The main demographic and clinical characteristics are displayed in Table 1. All ARVC patients were tested with a broad cardiomyopathy gene panel. All except two had a pathogenic mutation in one of the desmosomal genes involved in ARVC pathogenesis (Table 1). The remaining two patients had no known mutations nor variants of unknown significance in any cardiomyopathy gene. All M-carriers and controls were tested for the family desmosomal mutation alone.

### Location and timing of terminal ventricular activation area

**Controls.** The $aVAte_{20}$ was located in the RVOT and the LV base (+/- RV base) in all controls (Fig 1) and also in parts of the LV apex and LV lateral wall in three.

**ARVC patients.** The $aVAte_{20}$ was restricted to the RV alone in all patients. The $aVAte_{20}$ was located in the RVOT in 8 cases (67%), RV base in 5 cases (42%), RVIW in 3 cases (25%), RV apex in 2 cases (17%) and in the RV free wall in 1 case (8%). The regions involved during the terminal ventricular activation were either one isolated or multiple isolated locations (Fig 2). The mean $aVAte_{20}$ was significantly smaller in ARVC patients compared with controls (35 cm$^2$ vs 87 cm$^2$, p<0.05). Both tVAT and $TaVAte_{20}$ were significantly longer in ARVC

**Table 1. Baseline clinical characteristics.**

|  | Controls (n = 8) | ARVC patients (n = 12) | M-carriers (n = 20) |
|---|---|---|---|
| Sex, males | 5 (62.5) | 8 (66.7) | 8 (40) |
| Age, years, mean (SD) | 39 (18) | 50 (16) | 44 (14) |
| Desmosomal mutations* | 0 (0) | 10 (83) | 20 (100) |
| Clinical events |  |  |  |
| Cardiac syncope | 0 | 3 (25) | 0 |
| Aborted SCD | 0 | 1 (8) | 0 |
| Non-sustained VT | 0 | 4 (33) | 0 |
| Sustained VT | 0 | 7 (58) | 0 |
| ≥500 VES/ 24hours | 0 | 8 (67) | 0 |
| ICD | 0 | 10 (83) | 0 |
| Antiarrhythmic drugs | 0 | 4 (33) | 0 |
| Prior VT ablation | 0 | 1 (8) | 0 |
| RV-FAC <40% | 0 | 11 (92) | 0 |
| LV-EF % (mean ±SD) | 62 (4) | 63 (4) | 61 (4) |
| 12 lead ECG |  |  |  |
| T wave inversion V1-V3 or beyond | 0 | 10 (83) | 2 (10) |
| Prolonged TAD in V1, V2 or V3 | 0 | 8 (67) | 2 (10) |
| Epsilon waves | 0 | 6 (50) | 0 |
| Late potentials (1–3 criteria) on SAECG: | 2 (25) | 11 (92) | 5 (25) |

The figures are numbers with percentage in brackets, unless stated otherwise.

*Prevalence of desmosomal mutations in ARVC patients: PKP-2 gene (58%), DSP gene (8%), DSG-2 gene (16%). Familial mutations in M-carriers: PKP-2 gene (50%), DSP gene (20%), DSG-2 gene (15%), DSC-2 gene (15%).

n = number of study subjects; PKP-2: plakophillin-2, DSP: desmoplakin, DSG-2: desmoglein-2, DSC-2: desmocollin-2, SCD = Sudden Cardiac Death; SD = standard deviation; VT = Ventricular Tachycardia; VES = Ventricular Extrasystole; ICD = Implantable Cardioverter Defibrillator; RV = right ventricle; RV-FAC: right ventricular–fractional area change; LV-EF = left ventricular ejection fraction; TAD = Terminal Activation Duration; SAECG = Signal Averaged-ECG.

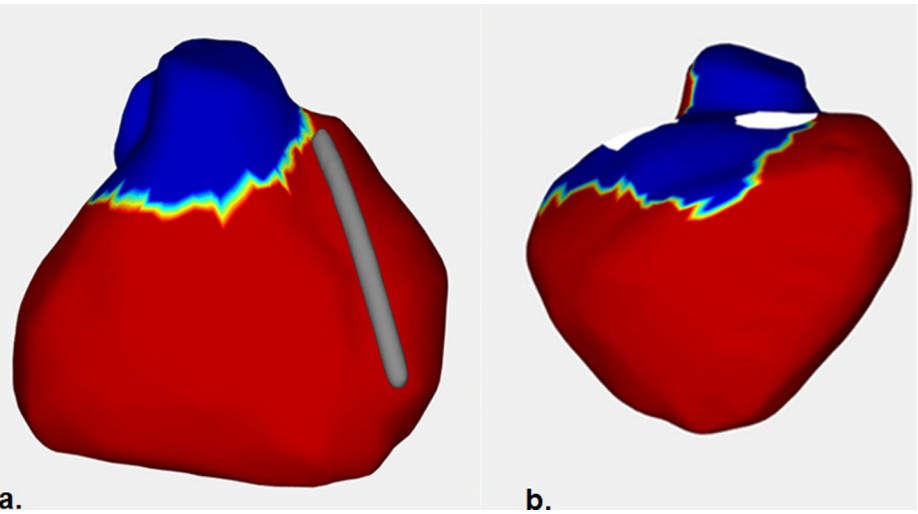

**Fig 1. Example of the pattern of terminal ventricular activation in a control subject.** The figure presents the area activated during the last 20 msec of the total ventricular epicardial activation (noted with blue color) in a control. Note that the activation during the last 20msec involves the RVOT (a) and parts of LV and RV base (b). a = RV anterior wall, b = RV inferior wall. ARVC = Arrhythmogenic Right Ventricular Cardiomyopathy, RVOT = right ventricular outflow tract, RV = right ventricle, LV = left ventricle.

patients compared with controls (99 msec vs 58 msec, p<0.05 and 92 msec vs 49 msec, p<0.05, respectively) (Table 2).

**M-carriers.** The $aVAte_{20}$ was located in the RVOT and the LV base alone, as seen in controls, in 10 cases (50%) (Fig 3A and 3B). The $aVAte_{20}$ was located in areas beyond RVOT and LV base, in 9 (45%) M-carriers; in the inferior wall in 6 cases (30%), the posterior in 1 case (5%), the RV apex in 1 case (5%) and anterior wall in 1 case (5%) (Fig 3C). The $aVAte_{20}$ involved the RVOT and RV inferior wall alone, but not LV base, in 1 case (5%) (Fig 3D).

When comparing conduction abnormalities recorded by the ECGI and the SAECG, 7 (70%) of the 10 M-carriers with abnormal $aVAte_{20}$ activation patterns had no abnormal depolarisation criteria on any ECG recording. Of the 5 M-carriers in total with late potentials on SAECG, the 3 who fulfilled all 3 criteria, had an abnormal $aVAte_{20}$ pattern, i.e. regional late activations in segments of RV whereas 2 had a normal $aVAte_{20}$ pattern.

The mean $aVAte_{20}$ in M-carriers was similar to those found in controls (70 cm$^2$ vs 87 cm$^2$, p = 0.070) (Table 2). The tVAT and $TaVAte_{20}$ were also similar to those in controls (63 msec vs 58 msec, p = 0.328; 52 msec vs 49 msec, p = 0.566) (Table 2). The sub-group with eccentric $aVAte_{20}$, however, had significantly smaller $aVAte_{20}$ as compared with the M-carriers with normally distributed and activated $aVAte_{20}$ (53cm$^2$ vs 87cm$^2$, p = 0.009). Both tVAT and $TaVAte_{20}$, however, did not differ in these two M-carrier sub-groups (65msec vs 60msec, p = 0.529, 54msec vs 50msec, p = 0.579).

## Right Ventricular Activation patterns and durations

**Controls.** The epicardial activation of the RV started in the paraseptal region and was completed at the RVOT in 6 cases (75%). In one case the activation started at the apical segment of RVIW and ended in the RVOT while in another it started in the RV anteroparaseptal region and ended in the basal segment of RVIW.

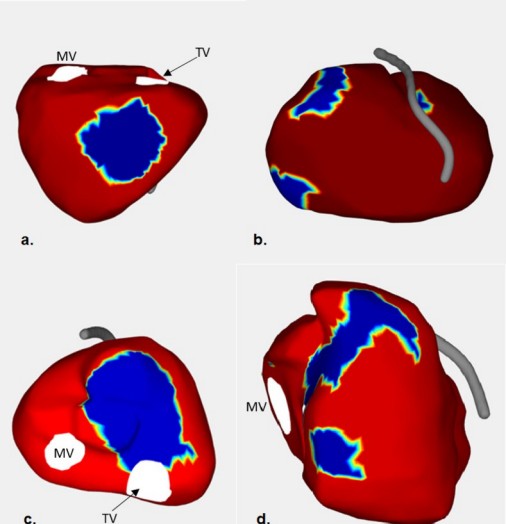

**Fig 2. Examples of terminal ventricular activation in four ARVC patients.** The figure presents the area activated during the last 20 msec of the total ventricular epicardial activation (noted with blue colour) in four ARVC patients. Note that the terminal ventricular activation varied among ARVC patients and occurred in the RV inferior wall (a); in three different areas of the RV (paraseptal, RV base and RV free wall)(b); isolated in RVOT (c);and in two separate areas in RVOT and RV base (d). ARVC = Arrhythmogenic Right Ventricular Cardiomyopathy, RVOT = right ventricular outflow tract, RV = right ventricle, LV = left ventricle, TV: tricuspidalis valve, MV: mitralis valve, LAD = left anterior descending coronary artery.

**Table 2. Characteristics of epicardial ventricular activation.**

| | Controls (n = 8) | ARVC patients (n = 12) | M-carriers (n = 20) | p-values* |
|---|---|---|---|---|
| Terminal ventricular activation | | | | |
| aVAte$_{20}$ (cm$^2$) | 87 (19) | 35 (27) | 70 (35) | <0.05 |
| tVAT (msec) | 58 (11) | 99 (21) | 63 (9) | <0.05 |
| TaVAte$_{20}$ (msec) | 49 (11) | 92 (23) | 52 (9) | <0.05 |
| RV activation time | | | | |
| RVAT (msec) | 43 (7) | 66 (18) | 51 (9) | <0.05 |
| RVAT-AW (msec) | 43 (6) | 57 (17) | 49 (8) | 0.073 |
| RVAT-IW (msec) | 35 (8) | 47 (18) | 31 (12) | <0.05 |

The figures are mean values with one standard deviation in brackets, unless otherwise stated.

*The p values refer to the comparison between the three groups.

aVAte$_{20}$; the area of the terminal Ventricular Activation, tVAT = total Ventricular Activation Time, TaVAte$_{20}$ = mean time for the activation of all terminal areas,

RVAT: Right ventricular Activation Time, AW: anterior wall, IW: inferior wall, msec = milliseconds

**ARVC patients.** The ARVC group showed a greater variety for both earliest and latest activation locations but their location did not differ statistically from controls (Table 3). The earliest RV activation was in the right anteroparaseptal region and the latest in the RVOT in 5 cases (42%) of ARVC patients. The earliest RV activation was located in an abnormal segment (RV base, RVIW basal segment and RVAW) in 3 cases (25%) but the latest activation was still in the RVOT. The earliest activation was in the anteroparaseptal region and the latest activation in the basal segment of RVIW in 3 cases (25%) while the activation started at the RVIW apical segment and ended in the RV base in one case (Table 3).

**M-carriers.** Likewise, only in 8 cases (40%) of M-carriers was the earliest RV activation in the anteroparaseptal region and the latest in the RVOT as seen in controls. The earliest RV activation was in eccentric segments in 6 cases (30%) (RV base in 3, RVIW basal segment in 2 and RVAW in one case) but with the latest activation still in RVOT, as seen in controls. The

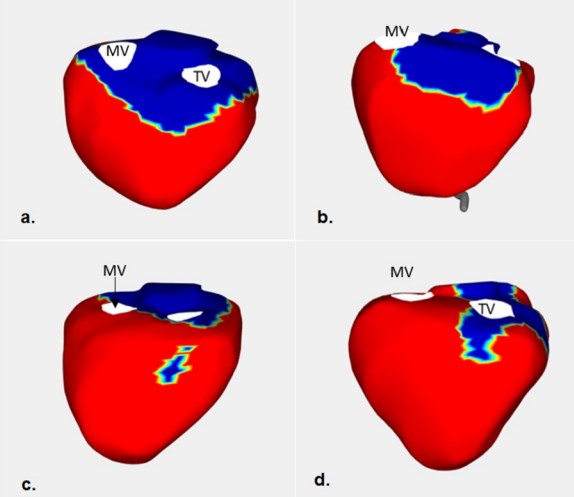

**Fig 3. Examples of terminal ventricular activation in four M-carriers.** The figure presents the area activated during the last 20 msec of the total ventricular epicardial activation (noted with blue colour) in four M-carriers. Note that the terminal ventricular activation could occur at RVOT and LV base, as in controls (a, b); could additionally involve other parts of the RV, like the inferior wall (c); the terminal ventricular activation could only involve parts of the RV, not involving the LV (d). Same abbreviations as in Fig 2.

**Table 3. Location of the earliest and latest activation of the RV.**

| | Controls (n = 8) | ARVC patients (n = 12) | M-carriers (n = 20) | p-values |
|---|---|---|---|---|
| Earliest activation whole RV | | | | 0.703 |
| Earliest activation RVAW | | | | |
| paraseptal | 87.5 | 66.7 | 65.0 | |
| base RVAW | 0 | 8.3 | 15.0 | |
| free wall | 0 | 8.3 | 10.0 | |
| Earliest activation RVIW | | | | |
| apical segment IW | 12.5 | 8.3 | 0 | |
| basal segment IW | 0 | 8.3 | 10.0 | |
| Latest activation whole RV | | | | 0.500 |
| Latest activation RVAW | | | | |
| RVOT | 87.5 | 66.7 | 70.0 | |
| Base RVAW | 0 | 8.3 | 0 | |
| Latest activation RVIW | | | | |
| basal segment IW | 12.5 | 25.0 | 30.0 | |

The figures denote the percentage of study subjects who present earliest or latest activation at respective RV segment in each group. The p-values refer to the comparison between the three groups with Kruskal-Wallis test.

RV: right ventricle, AW: anterior wall, IW: inferior wall, RVOT: right ventricle outflow tract, AT: activation time

earliest activation was in the anteroparaseptal region and the latest activation in the basal segment of RVIW in 5 cases (25%) while in one case the activation started at the free wall of RV and ended in the RVIW.

The RVAT and the RVAT-IW were both significantly longer among ARVC patients compared with controls (66 msec vs 43 msec, p<0.05 and 47 msec vs 35 msec, p<0.05, respectively). The activation duration of the RVAW in ARVC patients tended to be longer than controls but did not reach statistical significance (Table 2). When dividing the M-carriers in those with normal terminal ventricular activation and those with abnormal regions activated, the RVAT was longer among the latter sub-group (55msec vs 48msec, p = 0.043).

## RV epicardial depolarisation signal morphology

**Controls.** The RV epicardial depolarisation electrograms had an rS morphology in all regions except for in the RVOT and basal part of RVIW where it had an rSr´ pattern.

**ARVC patients.** The rS morphology was also the most dominant pattern in all RV segments except for in the RVOT and the basal part of RVIW, where an rSr´ morphology was dominant. The morphology of the epicardial electrograms showed a greater heterogeneity, particularly in the RV base and the RV apex, as compared with controls (Fig 4). Fractionated signals were seen in 7 ARVC patients (58%) as compared with none among controls.

**M-carriers.** The morphology of the epicardial signals was comparable to those in controls, i.e. rSr´ morphology in RVOT and basal RVIW and rS in the rest of the RV. Fractionated signals were observed in two cases.

## RV epicardial repolarization signal morphology

The T-waves were overall positive in *controls*, but were negative or isoelectric in the RVIW and the RV base in 3 and 6 controls, respectively.

The T-waves were negative, isoelectric or biphasic in RVOT and RVAW in 11 *ARVC cases* (92%) and in RVIW and RV base in all ARVC patients.

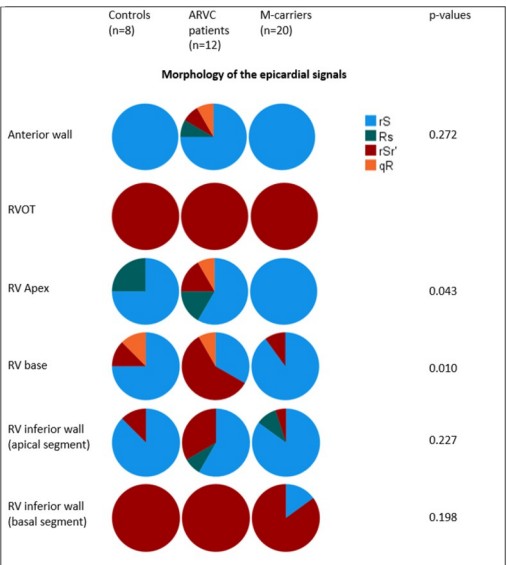

**Fig 4. Epicardial depolarisation signal morphology per study group and area of interest.** The figure summarizes the morphology of the epicardial QRS complexes, per study group and area of interest. The size of the pie chart reflects its proportional size. The p-values refer to the comparison between the three study groups with Kruskal-Wallis test. RV = right ventricle, RVOT = right ventricle outflow tract.

The T-waves were negative or isoelectric in RVOT or RVAW in 9 *M-carriers (45%)*, and in RVIW and RV base in 16 (80%) and 15 cases (75%) respectively (Fig 5). The 12-lead resting ECG showed T-wave inversions in only 2 cases, both of whom had epicardial T-wave inversions.

The data is summarized in S1 Table.

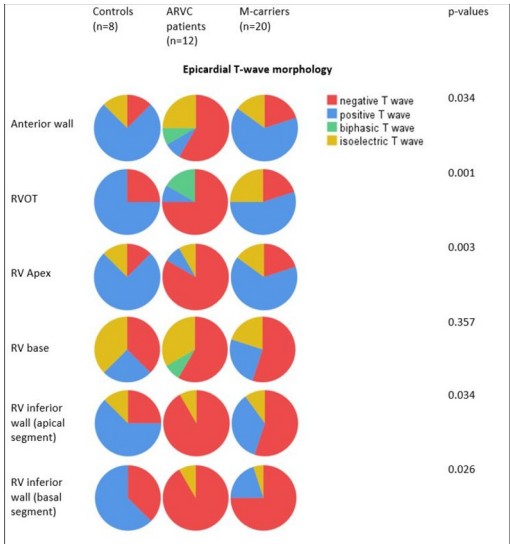

**Fig 5. Epicardial repolarisation morphology as reflected by T wave morphology per study group and area of interest.** The figure summarizes the morphology of the epicardial T-waves, per study group and area of interest. The size of the pie chart reflects its proportional size. The p-values refer to the comparison between the three study groups with Kruskal-Wallis test. RV = right ventricle, RVOT = right ventricle outflow tract.

## Discussion

The novel findings in the present study were the unmasking of conduction delays and repolarisation abnormalities in healthy M-carriers using ECGI. The finding may have important implications since M-carriers is an important and relatively large group that deserves more attention considering the importance of early detection of subclinical manifestations of the disease to prevent life-threating ventricular arrhythmias. Relatives in mutation negative ARVC families is an even more difficult group to manage and screen for during follow-ups. Although previous studies have used ECGI and found similar findings in ARVC patients regarding delayed conduction, fractionated signals and repolarisation abnormalities, none of them included M-carriers with no signs of the disease [7, 8].

The important finding in the present study was that a high proportion of M-carriers had regional conduction delays in parts of the RV. The finding that such conduction delays were present in individuals without corresponding depolarisation changes, such as epsilon potentials, on 12-lead resting ECG or late potentials on SAECG, suggests that epicardial lesions develop earlier and/or that conventional ECG/SAECG are less sensitive for such changes. Moreover, the finding that the tVAT was not prolonged, as opposed to the RVAT, in individuals with late RV activations, suggests that it may reflect an early stage of the disease limited to delayed RV conduction alone that is still completed before the depolarisation of the LV. Such regional delayed epicardial conduction may be a useful measure for the unmasking of an early manifestation of the disease. Further, their presence may potentially also explain why some ARVC patients present with ventricular arrhythmias before any signs of structural abnormalities or even before ECG changes are apparent [16].

Another novel finding in the present study is the finding of earliest or latest RV activation at abnormal sites in more than half (60%) of M-carriers. The deviating earliest RV activations may reflect an intrinsic local delayed activation at the anteroparaseptal region, so that other segments depolarize first. Even though the latest RV depolarisation was more often (70%) in the RVOT as in controls, it does not contradict the presence of areas with delayed activations, as they may well reach complete depolarisation by the time the activation is completed in the RVOT, findings compatible with early stages of the disease.

The finding of delayed epicardial activation in RV regions remote from the RVOT and LV/RV base in ARVC patients when using ECGI, differentiated them all from controls, in whom the terminal ventricular activation was localized to the RVOT and LV base (+/- RV base). The observation that the terminal ventricular activation areas were smaller in ARVC patients than in controls reflects the delayed conduction in localized areas of the RV. The findings of prolonged tVAT and RVAT and abnormally located earliest and latest activation patterns among ARVC patients compared with controls, confirm previous reports using ECGI [7, 8]. The above mentioned findings are all signs of a delayed RV conduction which may prove useful as early markers of subclinical disease.

The observation that the morphology of the epicardial signals deviated from that in controls in almost half of ARVC patients (42%) in the present study was consistent with previous reports [14, 17]. Fractionated signals, reflecting delayed or abnormal depolarization, is a well described feature in ARVC [18] that was more common in ARVC patients but also found in two M-carriers. However, neither the morphology of the epicardial signals nor the presence of fractionated signals could adequately differentiate ARVC patients from controls and could therefore not be used as a diagnostic marker of the disease.

The T-wave morphology of epicardial signals has not been described previously. The observation of T-wave changes in the ARVC group in the present study is a novel finding. The finding of a higher prevalence of negative/ isoelectric T-waves in epicardial signals than in surface

ECG signals among M-carriers may suggest that ECGI is more sensitive than conventional ECG in detecting repolarisation changes as well.

Early electrical disturbances in ARVC have been reported by several studies [19–21]. Murine and human studies have shown that delayed ventricular conduction detected by electrophysiological study precedes structural changes in ARVC [19]. Clinical study have shown that dynamic electrophysiological changes, such as conduction delay during ventricular pacing, particularly in short coupling intervals and repolarisation dispersions, can be revealed even in early stages of the disease [20]. Similarly, clinical study on relatives has shown that more relatives fulfil electrical criteria than imaging criteria for ARVC suggesting that diagnostic tools for visualization of electrical abnormalities may be of greater diagnostic value than current imaging tools visualizing morphological RV abnormalities [21].

Early conduction and repolarisation abnormalities in ARVC can be explained by the mutated desmosomes. Experimental studies have proven that the integrity of the electrical coupling between the myocytes depends on the integrity of the intercalated discs [22] and that it is disturbed due to mutations on desmosomal genes and the encoding of dysfunctional desmosomal proteins in ARVC patients [23–27]. Such abnormalities on the cellular level may result in an electrical instability of the myocardium that predisposes to the development of an arrhythmogenic phenotype, before any structural abnormalities can be detected macroscopically [28]. Thus the detection of electrophysiological disturbances can unmask the presence of an early manifestation of the disease, which can be used when screening both mutation carriers and relatives to families without a known mutation.

The potential role of invasive electrophysiologic procedure for diagnostic purposes in ARVC has been evaluated in several studies [29–33]. Low voltage areas and fragmented electrograms detected by electroanatomic mapping have been reported as common findings in ARVC patients that can enhance the diagnosis [18, 29, 30, 33]. Low-voltage areas, recorded from both the endocardium and epicardium, and RV endocardial activation delay have been reported as sensitive markers that can distinguish of ARVC from benign RVOT ventricular tachycardia [5, 34, 35]. Although an electrophysiologic procedure with electroanatomic mapping could be useful for diagnostic purposes its invasive nature and potential risk for complications makes it less suitable for serial evaluations.

ECGI is on the contrary a non-invasive technique without these limitations. The present study suggests that both depolarisation and repolarisation abnormalities can be detected and unmasked by ECGI at an early stage of the disease with a potential role for an early diagnosis.

The main limitation with the present technology of the body surface mapping vest, is its high costs which currently limits a more widespread application with serial investigations during follow-ups. Another limitation of the current study was that cardiac Magnetic Resonance Imaging (MRI) was not included in study protocoll. Even though electrical abnormalities have been reported to precede structural changes detected by MRI or echocardiography [36] and screening for electrical abnormalities is considered more sensitive than imaging modalities [21], one cannot totally exclude that MRI may have detected signs of the disease in a smaller portion of gene carriers. The study cohort was also limited; larger studies are warranted in order to confirm these results.

## Conclusion

The novel finding of localized delayed epicardial RV activations in genetically predisposed relatives without an ARVC phenotype indicates that ECGI may potentially have a diagnostic role in detecting early signs of the disease not yet unmasked by conventional diagnostic techniques.

The finding may be of great importance as it can enhance an early diagnosis and potentially be of value for risk stratification of these individuals.

## Supporting information

**S1 Table. Summary of the repolarization and depolarization characteristics of the cohort.** The table summarizes the main findings of the study regarding repolarization and depolarization and compares them with the findings from a 12-lead ECG. (Tw: T-wave, TwI: T-wave inversion, EW: epsilon-wave, TAD: terminal activation duration, LP: late potential, AW: anterior wall, IWa: apical segment of inferior wall, IWb: basal segment of inferior wall, P: positive, N: negative, Iso: isoelectric, Biph: biphasic, TA: terminal activation, ms: milliseconds). *Refers to the number of LP criteria that are fulfilled.
(DOCX)

## Author Contributions

**Conceptualization:** Varvara Kommata, Carina Blomström-Lundqvist.

**Data curation:** Varvara Kommata.

**Formal analysis:** Varvara Kommata.

**Funding acquisition:** Carina Blomström-Lundqvist.

**Investigation:** Varvara Kommata.

**Methodology:** Varvara Kommata, Carina Blomström-Lundqvist.

**Project administration:** Varvara Kommata, Carina Blomström-Lundqvist.

**Supervision:** Elena Sciaraffia, Carina Blomström-Lundqvist.

**Validation:** Carina Blomström-Lundqvist.

**Visualization:** Varvara Kommata.

**Writing – original draft:** Varvara Kommata.

**Writing – review & editing:** Varvara Kommata, Elena Sciaraffia, Carina Blomström-Lundqvist.

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
