## [Decision Letter · Decision Letter 0]

7 Oct 2022

PONE-D-22-21921Epicardial conduction abnormalities in patients with Arrhythmogenic Right Ventricular Cardiomyopathy (ARVC) and mutation positive healthy family members – a study using electrocardiographic imaging.PLOS ONE

Dear Dr. Kommata,

Thank you for submitting your manuscript to PLOS ONE. After careful consideration, we feel that it has merit but does not fully meet PLOS ONE’s publication criteria as it currently stands. Therefore, we invite you to submit a revised version of the manuscript that addresses the points raised during the review process.

We look forward to receiving your revised manuscript.

Kind regards,

Michela Casella, MD PhD

Academic Editor

PLOS ONE

Journal Requirements:

"The study received unrestricted grants from the Swedish Heart-Lung Foundation, Medtronic and Selanders Stiftelse. No sponsor had any role in the design and conduct of the study; collection, management, analysis, and interpretation of the data; preparation, review, or approval of the manuscript; and decision to submit the manuscript for publication."

"Varvara Kommata, Elena Sciaraffia have no conflict of interest to disclose. Carina Blomström-Lundqvist reports receiving grants from Medtronic during the conduct of the study; and personal fees from Bayer, Sanofi, Boston Scientific, and Cathprint outside the submitted work."

Reviewers' comments:

Reviewer's Responses to Questions

**Comments to the Author**

1. Is the manuscript technically sound, and do the data support the conclusions?

Reviewer #1: Yes

Reviewer #2: Partly

2. Has the statistical analysis been performed appropriately and rigorously? 

Reviewer #1: No

Reviewer #2: Yes

3. Have the authors made all data underlying the findings in their manuscript fully available?

Reviewer #1: Yes

Reviewer #2: No

4. Is the manuscript presented in an intelligible fashion and written in standard English?

Reviewer #1: Yes

Reviewer #2: Yes

5. Review Comments to the Author

Reviewer #1: Dear Authors,

I recently had the please of reviewing the manuscript entitled "Epicardial conduction abnormalities in patients with Arrhythmogenic 1 Right Ventricular Cardiomyopathy (ARVC) and mutation positive healthy family members – a study using electrocardiographic imaging", submitted to PLOS One by Dr. Kommata.

I perceive this manuscript as of great interest for the field of ARVC research. Many of the insights reported are of great value. These findings shed additional lighting on potential mechanist processes behind this disease and could potentially have implications at a risk stratification level. I believe the authors should be commended for their design study and for the presentation.

I have a few comments that I'd like to see addressed:

Methods:

1. Statistical analysis: p values in multiple tables are reported as overall p values. I believe this to be appropriate. In the result section, however, single pair-wise comparisons are performed. It is unclear to me if the p value reported for the pair-wise comparison is corrected for a post-hoc analysis or not. I would require additional specifics on this point.

2. I really like the comparisons between ARVC phenotype carriers, gene positive family members, and gene negative family members because I believe strengthens the reliability of data derived from this study.

One question here is this: I see three family members (15%, as per caption in Table1) having a DSC-2 mutation, while no ARVC phenotype carrier with that same mutation is reported. This seems counter intuitive to me, given the selection process for those patients. I would like the authors to elaborate on this.

3. This is a pet-peeve of mine but I'd recommend authors to report data in a consistent fashion.

When listing patients number, I'd recommend a notation  n (%).

That said, other notations are absolutely fine, but I'd recommend picking one and consistently use that through the text.

4. The findings from the CardioInsight analysis are fascinating.

I believe these findings may use additional clinical correlations.

- Do we have any association between the findings at the CardioInsight at the 12-lead ECG findings? i.e. do we have good agreement between localization area of T wave inversions, TAD, fractionation and findings at the CardioInsight analysis?

- Is there any correlation between epicardial conduction time and QRS duration?

- Any association between these findings and a) PVC burden; b) Arrhythmic outcomes at long term?

Reporting this data could really improve the clinical value of this study.

5. This reviewer fully understand the tyranny of small numbers, when doing research with relatively rare diseases such as ARVC.

That said, I would be really interested in seeing a sub-analysis grouping those patients & family members at a potential risk for a left-dominant disease or at least the non-PKP2 group (DSP, DSC-2 & DSG-2), to see if there is any relative differences in either localization of the area of delayed conduction or specific conduction times.

I fully understand this assessment could be underpowered. Nonetheless, I believe this sub-analysis significantly increasing the generalizability of this manuscript and I would strongly encourage the authors to present it (even in the supplementary materials).

6. Any comments on LV characteristics of those patients?

Best

Reviewer #2: Kommata and colleagues present a systematic assessment of electrocardiographic (ECG) imaging as a diagnostic tool for the detection of epicardial conduction abnormalities among 12 patients with ARVC, 20 healthy mutation carriers, and 8 controls.

The main finding is that patients with ARVC have longer right ventricular activation times and a different pattern of epicardial activation compared to controls, while a similar difference did not emerge in the comparison between healthy mutation carriers and controls, although 50% of healthy mutation carriers had similar findings to ARVC patients. The authors conclude that "the observation of localized delayed RV epicardial conduction in M-carriers suggests an early stage of ARVC and may be a useful diagnostic marker enhancing an early detection of the disease."

Although the paper is well written and presents an interesting assessment of a relatively novel technology, there are several major limitations that should be highlighted:

1) According to this reviewer, the abstract conclusion is not supported by the results. In fact, in order to identify a novel tool for ARVC diagnosis, the authors should provide evidence that findings from ECG imaging are linked to outcomes, especially to the occurrence of sustained ventricular tachycardia and/or ventricular fibrillation. This follow-up information would prove instrumental in the subset of healthy mutation carriers with abnormal ECG imaging findings, in whom stricter physical exercise restrictions would be recommended and closer clinic appointments/Holter monitoring scheduled.

2) The diagnostic assessment did not include CMR. Given the key role of CMR in disclosing subtle structural right ventricular abnormalities which could otherwise be missed, its superior accuracy compared to transthoracic echocardiography, and the renewed interest in LGE as a diagnostic criterion for ARVC (See Corrado et al, JAHA, 2021), this reviewer believes that any new diagnostic tool should be assessed on a baseline of all available diagnostic techniques including CMR. The finding that some mutation carriers had abnormal ECG imaging findings may suggest that some structural abnormalities were already present and could have been detected by CMR.

3) There are similar reports from the same group (See Kommata et al, PACE, 2022).

6. PLOS authors have the option to publish the peer review history of their article (what does this mean?). If published, this will include your full peer review and any attached files.

Reviewer #1: **Yes: **Alessio Gasperetti

Reviewer #2: No

---

## [Author Response · Author response to Decision Letter 0]

4 Dec 2022

Answer to the reviewers

Reviewer 1:

1. Statistical analysis: p values in multiple tables are reported as overall p values. I believe this to be appropriate. In the result section, however, single pair-wise comparisons are performed. It is unclear to me if the p value reported for the pair-wise comparison is corrected for a post-hoc analysis or not. I would require additional specifics on this point.

Thank you for noticing that! We indeed performed some single pair-wise analyses to further explore our data and identify potential variables of interest. Post-hoc analysis (bonferroni) was performed when appropriate. This is now specified on statistical analysis paragraph.

2. I really like the comparisons between ARVC phenotype carriers, gene positive family members, and gene negative family members because I believe strengthens the reliability of data derived from this study. One question here is this: I see three family members (15%, as per caption in Table1) having a DSC-2 mutation, while no ARVC phenotype carrier with that same mutation is reported. This seems counter intuitive to me, given the selection process for those patients. I would like the authors to elaborate on this.

Thank you for this remark and for giving us the opportunity to clarify that. As this was an explorative one-center study with a limited number of participants, a mutation-based analysis was out of the scope. In general, we aimed to include ARVC patients with a typical phenotype, predominant RV involvement and a broad genotypic spectrum. Unfortunately no ARVC patients with DSC-2 mutation were found eligible to participate, even though gene carriers of such mutations were included. In a future larger study, a more equally divided number of participants per genotype would be desirable.

3. This is a pet-peeve of mine but I'd recommend authors to report data in a consistent fashion.

When listing patients number, I'd recommend a notation  n (%). That said, other notations are absolutely fine, but I'd recommend picking one and consistently use that through the text.

Thank you for pointing out that. Now we have adjusted throughout the text.

4. The findings from the CardioInsight analysis are fascinating. I believe these findings may use additional clinical correlations.

- Do we have any association between the findings at the CardioInsight at the 12-lead ECG findings? i.e. do we have good agreement between localization area of T wave inversions, TAD, fractionation and findings at the CardioInsight analysis?

Thank you for this question which is very interesting. The assessment of ECGI findings and comparison with ECG findings in individual level was outside the scope of this manuscript, where we focused on the epicardial conduction as assessed by ECGI. 

We have summarized such comparisons in one table that we suggest can go to the supplement (S1 Table) and added reference to this table in the results section. 

- Is there any correlation between epicardial conduction time and QRS duration?

A Pearson correlation coefficient was computed to assess the linear relationship between QRS duration and epicardial conduction time (tVAT and RVAT). There was no linear correlation between these variables (tVAT: r=0.064, p=0.697, RVAT: r=0.002, p=0.989).

- Any association between these findings and a) PVC burden; b) Arrhythmic outcomes at long term? 

We think that it would be the next step to use these findings as prognostic markers of the disease but since the material is small as is the event rate, we focused on the diagnostic capability of ECGI at the present time. We plan to further analyze these findings in relation to arrhythmia burden in order to answer this important clinical question. 

Reporting this data could really improve the clinical value of this study.

5. This reviewer fully understand the tyranny of small numbers, when doing research with relatively rare diseases such as ARVC. That said, I would be really interested in seeing a sub-analysis grouping those patients & family members at a potential risk for a left-dominant disease or at least the non-PKP2 group (DSP, DSC-2 & DSG-2), to see if there is any relative differences in either localization of the area of delayed conduction or specific conduction times.

I fully understand this assessment could be underpowered. Nonetheless, I believe this sub-analysis significantly increasing the generalizability of this manuscript and I would strongly encourage the authors to present it (even in the supplementary materials). 

We agree with the reviewer that a gene-specific analysis as proposed would be very interesting but we abandoned that idea from the very start due to the small number of patients in our cohort. 

Nevertheless, a Mann-Whitney test revealed no statistically difference between gene carriers with a PKP-2 mutation and those with a mutation in another desmosomal gene, regarding aVAte20 (63.6 cm2 vs 75.8 cm2, p=0.579), TaVAte20 (54msec vs 49.4msec, p=0.529) and tVAT (65.5msec vs 60.1msec, p=0.280). Pearson Chi-Square test revealed no difference on the pattern of terminal ventricular activation (p=1.0) and location of area of delayed conduction (p=0.478). No difference was revealed in ARVC group neither after Kruskal-Wallis test regarding aVAte20 (p=0.690), TaVAte20 (p=0.608) and tVAT (p=0.566). The pattern of terminal ventricular activation was pathological in all ARVC patients. Pearson Chi-Square test revealed no difference on the location of area of delayed conduction (p=0.164).

We do not think that the paragraph above adds value to the present manuscript. But if the reviewers insist, we could add this information on supplements.

6. Any comments on LV characteristics of those patients?

In this study, which is the first study exploring systematically a new implementation of ECGI in ARVC diagnostics during sinus rhythm, we have only included ARVC patients with a typical phenotype, i.e. a primarily RV involvement. No patients had any echocardiographic signs of left ventricular dysfunction in this study. The epicardial conduction of the LV has not yet been investigated. This should be very interesting to further investigate in future projects including patients with a more variable phenotype. 

Reviewer 2:

1) According to this reviewer, the abstract conclusion is not supported by the results. In fact, in order to identify a novel tool for ARVC diagnosis, the authors should provide evidence that findings from ECG imaging are linked to outcomes, especially to the occurrence of sustained ventricular tachycardia and/or ventricular fibrillation. This follow-up information would prove instrumental in the subset of healthy mutation carriers with abnormal ECG imaging findings, in whom stricter physical exercise restrictions would be recommended and closer clinic appointments/Holter monitoring scheduled.

We appreciate the reviewer’s comment. As we have emphasized in our manuscript, in this study we focused on the potential role of ECGI to improve ARVC diagnosis, by identifying potential markers of abnormal epicardial conduction. This is completely different from its application as a prognostic marker for ventricular arrhythmias which was beyond the scope of this study. We have indeed planned another study exploring the role of ECGI as a predictor for the occurrence and/or burden of ventricular arrhythmias. 

2) The diagnostic assessment did not include CMR. Given the key role of CMR in disclosing subtle structural right ventricular abnormalities which could otherwise be missed, its superior accuracy compared to transthoracic echocardiography, and the renewed interest in LGE as a diagnostic criterion for ARVC (See Corrado et al, JAHA, 2021), this reviewer believes that any new diagnostic tool should be assessed on a baseline of all available diagnostic techniques including CMR. The finding that some mutation carriers had abnormal ECG imaging findings may suggest that some structural abnormalities were already present and could have been detected by CMR. 

Thank you for this very important comment. In this explorative study, MRI was unfortunately not included in the study protocol. We agree with the reviewer that we cannot exclude that MRI may have detected signs of the disease in a smaller portion of gene carriers. It has, however, been shown that electrical abnormalities precede detectable structural changes as shown by MRI or echo (1). Moreover, screening for electrical abnormalities including 12 lead ECG, signal-averaged ECG, and Holter monitoring has been reported to be more sensitive than imaging modalities(2). 

We have explained this more clearly in the text at the end of the discussion: “Another limitation of the current study was that cardiac Magnetic Resonance Imaging (MRI) was not included in study protocoll. Even though electrical abnormalities have been reported to precede structural changes detected by MRI or echocardiography, and screening for electrical abnormalities is considered more sensitive than imaging modalities, one cannot totally exclude that MRI may have detected signs of the disease in a smaller portion of gene carriers.”

3) There are similar reports from the same group (See Kommata et al, PACE, 2022).

Thank you for paying attention to our previous work. These data were based on the analysis of the recorded body-surface signals. The present study focused on the reconstructed epicardial signals for the identification of delayed epicardial conduction. The signals investigated in this publication are of totally different nature/ character and the methodology of our analyses, as well as our results are totally different. 

 

References

1. te Riele AS, James CA, Rastegar N, Bhonsale A, Murray B, Tichnell C, et al. Yield of serial evaluation in at-risk family members of patients with ARVD/C. J Am Coll Cardiol. 2014;64(3):293-301.

2. Jurlander R, Mills HL, Espersen KI, Raja AA, Svendsen JH, Theilade J, et al. Screening relatives in arrhythmogenic right ventricular cardiomyopathy: yield of imaging and electrical investigations. Eur Heart J Cardiovasc Imaging. 2020;21(2):175-82.

 

Journal Requirements:

The text and the files are now adjusted.

"The study received unrestricted grants from the Swedish Heart-Lung Foundation, Medtronic and Selanders Stiftelse. No sponsor had any role in the design and conduct of the study; collection, management, analysis, and interpretation of the data; preparation, review, or approval of the manuscript; and decision to submit the manuscript for publication."

This is now adjusted in the text (marked with red).

"Varvara Kommata, Elena Sciaraffia have no conflict of interest to disclose. Carina Blomström-Lundqvist reports receiving grants from Medtronic during the conduct of the study; and personal fees from Bayer, Sanofi, Boston Scientific, and Cathprint outside the submitted work."

This is now addressed in our disclosures as following: “This does not alter our adherence to PLOS ONE policies on sharing data and materials. Due to ethical considerations, though, the data are available upon request.”

This is now added at the cover letter

---

## [Editor Report · Decision Letter 1]

20 Dec 2022

Epicardial conduction abnormalities in patients with Arrhythmogenic Right Ventricular Cardiomyopathy (ARVC) and mutation positive healthy family members – a study using electrocardiographic imaging.

PONE-D-22-21921R1

Dear Dr. Kommata,

We’re pleased to inform you that your manuscript has been judged scientifically suitable for publication and will be formally accepted for publication once it meets all outstanding technical requirements.

Kind regards,

Michela Casella, MD PhD

Academic Editor

PLOS ONE
---

## [Editor Report · Acceptance letter]

26 Dec 2022

PONE-D-22-21921R1 

Epicardial conduction abnormalities in patients with Arrhythmogenic Right Ventricular Cardiomyopathy (ARVC) and mutation positive healthy family members – a study using electrocardiographic imaging. 

Dear Dr. Kommata:

I'm pleased to inform you that your manuscript has been deemed suitable for publication in PLOS ONE. Congratulations! Your manuscript is now with our production department. 

Kind regards, 

on behalf of

Dr. Michela Casella 

Academic Editor

PLOS ONE